# Key Components, Current Practice and Clinical Outcomes of ERAS Programs in Patients Undergoing Orthopedic Surgery: A Systematic Review

**DOI:** 10.3390/jcm11144222

**Published:** 2022-07-20

**Authors:** Francesca Salamanna, Deyanira Contartese, Silvia Brogini, Andrea Visani, Konstantinos Martikos, Cristiana Griffoni, Alessandro Ricci, Alessandro Gasbarrini, Milena Fini

**Affiliations:** 1Complex Structure Surgical Sciences and Technologies, IRCCS Istituto Ortopedico Rizzoli, 40136 Bologna, Italy; francesca.salamanna@ior.it (F.S.); deyanira.contartese@ior.it (D.C.); andrea.visani@ior.it (A.V.); milena.fini@ior.it (M.F.); 2Spine Surgery Unit, IRCCS Istituto Ortopedico Rizzoli, 40136 Bologna, Italy; konstantinos.martikos@ior.it (K.M.); cristiana.griffoni@ior.it (C.G.); alessandro.gasbarrini@ior.it (A.G.); 3Anesthesia-Resuscitation and Intensive Care, IRCCS Istituto Ortopedico Rizzoli, 40136 Bologna, Italy; alessandro.ricci@ior.it

**Keywords:** ERAS, orthopedic surgery, preoperative, perioperative, postoperative elements

## Abstract

Enhanced recovery after surgery (ERAS) protocols have led to improvements in outcomes in several surgical fields, through multimodal optimization of patient pathways, reductions in complications, improved patient experiences and reductions in the length of stay. However, their use has not been uniformly recognized in all orthopedic fields, and there is still no consensus on the best implementation process. Here, we evaluated pre-, peri-, and post-operative key elements and clinical evidence of ERAS protocols, measurements, and associated outcomes in patients undergoing different orthopedic surgical procedures. A systematic literature search on PubMed, Scopus, and Web of Science Core Collection databases was conducted to identify clinical studies, from 2012 to 2022. Out of the 1154 studies retrieved, 174 (25 on spine surgery, 4 on thorax surgery, 2 on elbow surgery and 143 on hip and/or knee surgery) were considered eligible for this review. Results showed that ERAS protocols improve the recovery from orthopedic surgery, decreasing the length of hospital stays (LOS) and the readmission rates. Comparative studies between ERAS and non-ERAS protocols also showed improvement in patient pain scores, satisfaction, and range of motion. Although ERAS protocols in orthopedic surgery are safe and effective, future studies focusing on specific ERAS elements, in particular for elbow, thorax and spine, are mandatory to optimize the protocols.

## 1. Introduction

### 1.1. ERAS in Orthopedic Surgery

Currently, orthopedic surgery remains one of the most common hospital surgeries in the world with an ever-growing burden in low- and middle-income countries. The number of orthopedic procedures performed worldwide totaled approximately 22.3 million in 2017 [1]. Additionally, the rising life expectancy in association with the shorter disease-free life expectancy (62.6 years in males and 64.4 years in females) will lead to an ever-increasing growth in the number of these procedures [2]. As demand for orthopedic surgical procedures has increased considering the recent advances in surgical and anesthesiologic techniques, the clinical pathways and care programs have undergone considerable changes influenced by the concept of ERAS programs [3]. ERAS aims to enhance the recovery from orthopedic surgery, also decreasing the length of hospital stays (LOS) and the readmission rates after surgery [3]. The reductions in LOS and readmission lead, in turn, to cost cutting and to a lower risk of nosocomial infections and thromboembolic events, as well as to a reduction in perioperative complications [3,4].

### 1.2. ERAS Protocols

ERAS protocols were introduced more than 20 years ago by Henrik Kehlet, providing the involvement of a multidisciplinary team made up of orthopedic surgeons, nursing staff, anesthesiologists, internists, physiatrists, physiotherapists, and nutritionists [5]. The procedures manage the patients’ care using a multi-modal approach that includes patient selection, patient-specific education and information on the preoperative, perioperative, and postoperative steps, improvements in surgical and anesthetic practices, advances in post-operative multi-modal analgesia, early rehabilitation and ambulation, early nutrition hydration, and discharge within 24 h post-surgery [6,7]. Preoperative patient education is of key importance in orthopedic care programs, particularly in ERAS programs, although its real impact with respect to traditional (standard) care in terms of anxiety, postoperative pain management, function, quality of life and complications is not yet clear [8,9]. Nevertheless, several studies recognized that satisfactory patient information is a critical element for early discharge and managing daily home life in ERAS programs, also supporting the value of multimodal education of the patient [10,11]. Additional key issues in ERAS programs in orthopedic surgery include effective pain treatment and management, which undoubtedly influence an early hospital discharge and a fast recovery period at home [10,12]. However, some studies evaluating the discharge procedure and patients’ experiences after hospital discharge showed that the early discharge, especially in elderly patients, may be stressful in terms of managing daily life and rehabilitation [10,11,12,13]. Although this type of ERAS pathway has undeniable advantages and represents the standard of care in many institutions, to date, the clinical effectiveness of ERAS procedures has not been homogeneously recognized or accepted for all orthopedic areas, and there is still significant work and research to be done [14,15,16]. In addition, the ERAS pathways are always undergoing improvement, thanks to the constant contribution that can derive from multiple perspectives such as that of the patient, the surgeon, or the hospital unit with the aim of improving the protocols. Continuous evidence-based revisions for ERAS use in different orthopedic areas are mandatory to properly update orthopedic surgeons and their staff on the use of these ERAS pathways and on their potential advantages over standard/traditional protocols in terms of safety and efficacy. In addition, within an optimized and clear ERAS protocol, selected high-risk patients may benefit from a planned longer stay in hospital as the best means of accelerating recovery and reducing complications, readmissions, and morbidity, and allowing the medical staff to monitor patients for longer periods of time. Thus, to highlight recent improvements in the preoperative, perioperative, and postoperative ERAS components and their clinical evidence in patients undergoing different types of orthopedic surgery, we carried out a systematic review to provide an evidenced-based assessment of specific interventions, measurement, and associated clinical outcomes linked to ERAS pathways in the orthopedic field.

## 2. Materials and Methods

### 2.1. Eligibility Criteria

The PICOS framework (population, intervention, comparison, outcomes, study design) [17] was used to formulate the questions for this study: (1) patients undergoing orthopedic surgery (population) submitted to, (2) ERAS pathways (interventions), (3) with or without a comparison group (standard protocol) (comparisons), (4) that reported preoperative, perioperative, and postoperative key components and clinical outcomes of the ERAS protocols (outcomes), in (5) randomized, non-randomized, controlled, non-controlled, retrospective, and prospective studies (study design). The focused question was “What are the preoperative, perioperative, and postoperative key components and the clinical outcomes of ERAS interventions in patients undergoing orthopedic surgery?”. Studies from 1 August 2011 to 1 August 2021, were included in this review if they met the PICOS criteria.

We excluded studies (1) in which the use of an ERAS protocol was declared but which then did not follow any of the indications of an ERAS protocol, and studies that evaluated (2) surgeries other than orthopedic ones, (3) patients undergoing orthopedic surgery with other concomitant severe pathological conditions (e.g., tumor, metastases, diabetes, rare neurological diseases, opioid use disorders), (4) different surgeries within a single ERAS protocol, (5) novel intervention/drugs/therapies not associated with ERAS protocols, and (6) articles with incorrect or incomplete data, or articles whose data could not be extracted. Additionally, we excluded abstracts, protocol studies, editorials, pilot studies, case reports or series, animal experiments, letters, comments to editors, reviews, meta-analyses, book chapters and articles not written in English.

### 2.2. Information Source and Search Strategies

Our literature review involved a systematic search conducted on 1 August 2021. We performed our review according to the Preferred Reporting Items for Systematic Reviews and Meta-Analyses (PRISMA) 2020 statement [18]. The search was carried out on PubMed, Scopus, and Web of Science Core Collection databases to identify studies that evaluated preoperative, perioperative, and postoperative key components and clinical evidence of ERAS protocols in orthopedic surgery. The search was conducted combining the terms (orthopedic disorders OR orthopedic surgery) AND (fast-track OR enhanced recovery after surgery OR enhanced recovery programs); for each of these terms, free words and controlled vocabulary specific to each bibliographic database were combined using the operator “OR”. The combination of free-vocabulary and/or medical subject headings (MeSH) terms for the identification of studies in PubMed, Scopus and Web of Science Core Collection are reported in Appendix A.

### 2.3. Selection Process

Possible relevant articles were screened using titles and abstracts by three reviewers (DC, FS, SB). After screening the titles and abstracts, articles were submitted to a public reference manager (Mendeley Desktop 1.19.8) to eliminate duplicates. Three reviewers (DC, FS, SB) performed 100% double title and abstract screening independently with inter-reviewer agreement of 90.1%. Studies that did not meet the inclusion criteria were excluded from full text review, and any disagreement was resolved through discussion until a consensus was reached, or with the involvement of a fourth reviewer (MF). Subsequently, the studies were subjected to full text review by three reviewers independently (DC, FS, SB). Disagreements after full text review were resolved through discussion, and the remaining studies were included in the final stage of data extraction. The inter-reviewer agreement for the final stage of data extraction was 95.3%.

### 2.4. Assessment of Methodological Quality

Two reviewers (DC, FS) independently assessed the methodological quality of selected studies. In case of disagreement, they attempted to reach consensus; if this failed, a third reviewer made the final decision (MF). The methodological quality of the studies was assessed using the quality assessment tools of the National Heart, Lung, and Blood Institute (NHLBI) [19] (Appendix A).

### 2.5. Data Collection Process and Synthesis Methods

The data extraction and synthesis process commenced with cataloguing the studies in detail. Subsequently, to increase validity and avoid potentially omitting findings for the synthesis, three authors (DC, FS, SB) extracted the data and generated tables taking into consideration the study design, pathological condition, patient numbers, ages and genders, surgical procedures, follow-up and outcomes/endpoints (Appendix A, Appendix A). Another table included ERAS protocols (preoperative, perioperative, postoperative) (Appendix A). Finally, a supplementary table (Appendix A) with a numerical designation of positive, neutral and negative outcomes for each study was reported.

## 3. Results

### 3.1. Study Selection and Characteristics

The initial literature search retrieved 1154 studies. Of those, 763 studies were identified using PubMed and 263 using Scopus, and 128 were found in the Web of Science Core Collection. Articles were submitted to a public reference manager to eliminate duplicate articles. The resulting 930 articles were screened for titles and abstracts, and 229 articles were then reviewed to establish whether the publication met the inclusion criteria. Finally, 174 (two on elbow orthopedic surgery, four on thorax orthopedic surgery, 25 on spine orthopedic surgery, and 143 on hip and/or knee orthopedic surgery, of which 52 were only on knee, 39 only on hip, and 52 on both knee and hip) were considered eligible for this review. Search research and study inclusion and exclusion criteria are detailed in Figure 1.

Of these articles, 82 were retrospective cohort studies (one on elbow, four on thorax, 18 on spine, 59 on knee and/or hip), 68 were prospective cohort studies (two of which were with a retrospective, historical cohort as control; six on spine, 62 on knee and/or hip) and 24 were randomized clinical trials (RCT) (one on elbow, one on spine, 22 on knee and/or hip) (Figure 2).

### 3.2. Assessment of Methodological Quality

The quality assessment for the two studies on orthopedic elbow surgery was strong for the single RCT and moderate for the one retrospective study, with weaknesses in the patient’s eligibility, sample size justification, blinded assessor, and potential confounding variables. Regarding the four studies on orthopedic thorax surgery, three studies were classified as moderate and one as weak, with weaknesses in, sample size justification, blinded assessor, and potential confounding variables examination. In the quality assessment of the 25 studies on orthopedic spine surgery, 12% of the studies were rated strong, 80% were rated moderate, and 8% were rated weak. Methodological weaknesses that led to study quality scores of moderate or weak often included the lack of a sample size justification and/or lack of variance and effect estimates, the lack of ERAS results evaluation more than once over time, the lack of blinded assessor and the lack of measurement of potential confounding variables. For the 143 studies on hip and/or knee orthopedic surgery, 39.2% were rated strong, 40.5% as moderate, and 20.3% as weak. The quality scores of moderate or weak studies included lack of a sample size justification and/or lack of variance and effect estimates, lack of ERAS results evaluation more than once over time, lack of blinded assessor and lack of measurement of potential confounding variables. Risks of bias assessments for each study are summarized in Appendix A.

### 3.3. Study Results and Synthesis

#### 3.3.1. Types of Orthopedic Surgery in ERAS Protocols

Of the 174 articles on ERAS selected and included in this review, 36.2% had a comparison with a standard/traditional protocol (non-ERAS), while all the others (64%) evaluated different ERAS protocols in patients undergoing orthopedic surgery. Of the 174 articles, 1.1% were on elbow orthopedic surgery, 2.3% on thorax orthopedic surgery, 14.4% on spine orthopedic surgery, and 82.2% on hip and/or knee orthopedic surgery. These data highlighted that the highest percentage of articles on ERAS were on total and mono-compartmental hip arthroplasty and knee arthroplasty, mainly performed due to osteoarthritis (OA) (62.2%) but, in some cases, also for fractures, avascular necrosis and revision surgery. However, it was also shown that ERAS programs are starting to apply to other orthopedic surgical specialties such as for spine, principally for spinal stenosis (36%), spinal scoliosis and deformities (32%) and adolescent idiopathic scoliosis (16%). Four articles on thorax orthopedic surgery were also present and used ERAS protocols for pectus deformities (n = 3) and for traumatic rib fracture (n = 1). Finally, two studies on ERAS protocols were present for patients with elbow post-traumatic stiffness and with elbow primary or secondary OA.

#### 3.3.2. Key Components in ERAS Protocols

##### Preoperative

Preoperative ERAS components are defined in this review as interventions that occur any time before the day of surgery, elements planned to optimize the patient’s condition prior to surgery. They also include advice about behavioral health and psychology referral to guide patients’ expectations as well as to inform them on the risks about intra- and postoperative pathways. Below are reported the preoperative ERAS components for the different orthopedic specialties (Figure 3).

-Elbow: In elbow orthopedic surgery, the most common preoperative interventions were patient education and the provision of information (on the surgical procedure, analgesia, anesthesia, LOS, and physiotherapy) (100%).-Thorax: The most common preoperative interventions reported in thorax orthopedic surgery were patient education and the provision of information (50%). Supplementary pre-emptive interventions were analgesia and multimodal pain management (50%) (defined as the use of one or more analgesic modes, such as acetaminophen, pregabalin, gabapentin, ketamine, non-steroidal anti-inflammatory drugs (NSAIDs), and cyclooxygenase (COX)-2 inhibitors)), clear fluid fasting (25%), physiotherapy (25%) and nausea and vomiting prophylaxis (25%).-Spine: In spine orthopedic surgery, among the principal interventions of ERAS protocol were patient education and the provision of information associated with a multidisciplinary consultation (geriatric, psychological, nutritional, behavioral health) (88%). Clear fluid and solid fluid fasting for 2–6 h before surgery (48%), pre-emptive analgesia and multimodal pain management (32%), antimicrobial/antibiotic prophylaxis (32%), nausea and vomiting prevention (20%), thromboprophylaxis (16%), tranexamic acid (TXA) (including oral or parenteral formulations) used to minimize bleeding (8%) and physiotherapy (4%) were other key interventions in spine orthopedic surgery.-Hip and/or knee: For hip and/or knee orthopedic surgery, the most common pre-operative interventions were patient education, the provision of information, and multidisciplinary consultation (43.3%), followed by pre-emptive analgesia and multimodal pain management (30%), comorbidities assessment (21.7%), antimicrobial/antibiotic prophylaxis (9.7%), clear fluid and solid fluid fasting for 2–6 h before surgery (8.3%), TXA use (7%) and thromboprophylaxis (4.9%).

##### Perioperative

Perioperative ERAS components/elements refer to all the interventions that occur from surgery until patient transfer to the post-anesthesia care unit (PACU). Below, and in Figure 3, are reported the perioperative ERAS components for the different orthopedic specialties (Figure 3).

-Elbow: In elbow orthopedic surgery frequent perioperative interventions were local anesthesia (50%), antimicrobial/antibiotic prophylaxis (50%), TXA use (50%) and avoidance of catheter/drain (50%).-Thorax: The most common perioperative interventions in thorax surgery included local anesthesia (50%), avoidance of catheter/drain (50%), antibiotic/antimicrobial prophylaxis (50%), fluid management (50%) and multimodal pain management (50%).-Spine: For spine surgery, perioperative components were multimodal analgesia and pain management (68%), local anesthesia (56%), normothermia/normovolemia maintenance (32%), TXA use (28%), antimicrobial/antibiotic prophylaxis (28%), postoperative nausea and vomiting prophylaxis (24%), transfusion control (20%) and avoidance of catheter/drain (20%).-Hip and/or knee: For hip and/or knee orthopedic surgery, the most common perioperative elements were local anesthesia (70%), multimodal pain management (55.2%), TXA use (36%), avoidance of catheter/drain (23%), intraoperative fluid management (14%), thromboprophylaxis (10.4%), compression bandage use (7%) and antimicrobial/antibiotic prophylaxis (5.6%).

##### Postoperative

Postoperative ERAS components are defined as interventions that occur during and after admission to the recovery area. Below, and in Figure 3, are described the postoperative ERAS components for the different orthopedic specialties (Figure 3).

-Elbow: The principal postoperative elements were early mobilization and rehabilitation/physiotherapy within 24 h (100% of studies) and multimodal analgesia and pain management (50%).-Thorax: In thorax surgery, key postoperative elements were represented by multimodal analgesia and pain management (75% of studies), early mobilization and rehabilitation/physiotherapy (50%), early nutrition (50%), catheter/drain removal within 24 h after surgery (25%) and nausea and vomiting prophylaxis (25%)-Spine: In spine surgery, postoperative elements were multimodal analgesia and pain management (84% of studies), early mobilization and rehabilitation/physiotherapy (64%), early nutrition (64%), catheter/drain removal within 24 h after surgery (32%), nausea and vomiting prophylaxis (12% in spine), thromboprophylaxis (12% spine), patient satisfaction survey (12%) and normothermia (4%).-Hip and/or knee: Principal postoperative elements were early mobilization and rehabilitation/physiotherapy (82% of studies), multimodal analgesia and pain management (61%), thromboprophylaxis (22.3%), early nutrition (7%), catheter/drain removal within 24 h after surgery (7%) and antimicrobial/antibiotic prophylaxis (5%).

### 3.4. Outcomes and Clinical Evidence of ERAS Protocols

All of the studies examined in this review confirmed the safety and efficacy of ERAS protocols in orthopedic surgery, showing an enhancement in the recovery from orthopedic surgery.

The primary outcomes in studies on elbow orthopedic surgery were a LOS reduction (50%), decrease in postoperative pain score (50%), especially in the first days after surgery, an abatement in drain removal time (50%), and an improvement in range of motion after ERAS pathway (50%).

Similarly, studies on thorax orthopedic surgery reported a significantly reduced LOS at 3 days after ERAS protocol, in patients undergoing minimally invasive repair of pectus excavatum (50%). Furthermore, a reduction in opioid consumption (50%), catheter removal time (50%), postoperative pain score (50%) and intraoperative time (25%), without an increase in the complication and readmission rate, was also noted after ERAS protocol.

In spine orthopedic surgery studies, a LOS of 1–3 days was observed for spinal deformities such as scoliosis and radiculopathy, while a LOS of 5–10 days was detected for lumbar stenosis or spondylolisthesis. Sixteen percent of studies also reported a significant reduction in intra-operative time after ERAS protocol. A reduction in catheter and drain removal time (12%), opioid consumption (12%), total health costs (16%), blood transfusion rate (8%), intraoperative blood loss (24%), postoperative pain score (24%), and complication and readmission rate (24%) were also detected in studies on spine orthopedic surgery. Finally, better functional recovery and early food recovery were observed in 20% and 12% of studies, respectively.

Concerning hip and/or knee orthopedic surgery, the most common reported outcomes were reductions in LOS (66.4%), postoperative pain score (25.2%), complication rate (16.8%) and bleeding rate/transfusion (13.3%), an increase in range of motion/walking anatomy/extension/flexion (13.3%), a reduction in readmission rate (9.8%) and opioid consumption (8.3%), a reduction in circulating markers of inflammation, anemia and endothelial activation (C-reactive protein, hemoglobin, tumor necrosis factor alpha) (8.3%), an increase in patient satisfaction (5.6%), and a reduction in intraoperative time (4.2%).

Almost all of the studies on elbow, thorax, spine and hip and/or knee orthopedic surgery that evaluated an ERAS vs. more conventional (non-fast track) (36%) protocol reported a significantly reduced LOS, without increasing complications or readmission rates in patients treated with ERAS regardless of follow-up (from 12 h to 5 years), surgical approach used, as well as surgeon. Only one study on spinal surgery did not find a significant change in LOS compared with the standard non-ERAS group [20]. In this study, an overall LOS increase, due to 5 h of observation in the PACU for a potential respiratory compromise, was detected. However, a variation in mean/median LOS, ranging from several hours to several days after surgery (from 12 h to 5.3 days), was observed between all the analyzed studies. Despite these variations, in all studies, the LOS reduction in the ERAS group was associated with a reduction in post-operative pain, bleeding rate and transfusion rate. The pain reduction during these ERAS pathways were associated with pre-emptive analgesia, perioperative local infiltration of analgesics (LIA) and post-operative analgesia. Several opioid-sparing agents were also used for pain relief in almost all studies. Specifically, paracetamol and NSAIDs were the most used. Analgesic protocols not only reduced the opioid requirements but also helped to reduce post-operative nausea-vomiting, post-operative stress, and the risk of complications. A reduction in transfusion rate with ERAS protocols vs. standard non-ERAS protocols was also seen in all of the studies that evaluated this element; this aspect was due not only to the optimization of hemoglobin mass performed in the preoperative phase but also to the prevention of perioperative blood loss. The main blood-saving strategy applied in this review was the TXA use. Depending on the study, TXA, an antifibrinolytic medication that stops the breakdown of fibrin clots by inhibiting activation of plasminogen, plasmin, and tissue plasminogen activator, was used in pre-, peri-, and post-operative phases. Several analyzed studies also evaluated different doses and administration routes (oral vs. intra-articular) of TXA, showing no differences with respect to blood loss and related thromboembolic events [21,22,23]. These ERAS elements not only improved the treatment management of the patients, increasing their satisfaction, but also aided the range of motion and return of function in all of the examined studies that evaluated these parameters (14.2%). Post-operatively, standard physiotherapy (kinesiotherapy) as well as other methods, including electrical stimulation, were also applied to strengthen the muscles, increase the range of motion, reduce swelling, and enhance independent gait, as it is known that early and persistent muscle loss occurs after these interventions, impairing balance and walking ability. The improvements in range of motion and return of function were undoubtedly helped by the early mobilization, but also by pain management as well as by the information and support given to the patients by the interdisciplinary team, because it increased their sense of self-efficacy, security, and satisfaction. Paradoxically, in their analysis 90 days after hip and knee arthroplasty, Jørgensen et al. found that fall-related hospital readmissions were due to physical activity and extrinsic factors other than surgery because of patient success and intent to return to a normal level of activity [24]. As emerged from all of the studies examined in this review, in turn, all of these interventions reduce the LOS as patients could be discharged sooner without increasing the risk of complications (References [25,26,27,28,29,30,31,32,33,34,35,36,37,38,39,40,41,42,43,44,45,46,47,48,49,50,51,52,53,54,55,56,57,58,59,60,61,62,63,64,65,66,67,68,69,70,71,72,73,74,75,76,77,78,79,80,81,82,83,84,85,86,87,88,89,90,91,92,93,94,95,96,97,98,99,100,101,102,103,104,105,106,107,108,109,110,111,112,113,114,115,116,117,118,119,120,121,122,123,124,125,126,127,128,129,130,131,132,133,134,135,136,137,138,139,140,141,142,143,144,145,146,147,148,149,150,151,152,153,154,155,156,157,158,159,160,161,162,163,164,165,166,167,168,169,170,171,172,173,174,175,176,177,178,179,180,181,182,183,184,185,186,187,188,189,190,191,192] are cited in the Appendix A).

## 4. Discussion

The ERAS philosophy focuses on patient experience, multidisciplinary teamwork (among surgeons, anesthesiologists, nurses, and physical therapists), evidence-based data gathering, and an iterative review process to improve protocol details across preoperative, perioperative, and postoperative phases [4,193]. Although the concept of ERAS was widely examined in orthopedic hip and/or knee replacement, its use in other orthopedic surgery has been employed only in recent years [194,195]. This aspect was specifically highlighted in this review where the presence of studies on ERAS in the elbow, thorax and spine emerged starting from 2018–2019, while numerous studies on ERAS in hip and/or knee replacement were present already in 2011. Although ERAS protocols seem to be well established and studied for specific orthopedic fields, this review highlighted the presence of numerous preliminary cohort studies lacking formal control groups (only 36.2% of the analyzed studies had a control group) and nonrandomized data sets as well as showing differences in postoperative follow-up, variability in operation and surgical indication in most of the studies, also for hip and/or knee replacement surgery [194].

A critical aspect that should be addressed with ERAS protocols would be to know which of the many elements really have an impact, thus, to understand if any of these elements may be skipped without resulting in inferior results, to further improve clinical outcomes and cost-efficacy of the protocol. However, it is important to underline that individual elements may not necessarily have significant benefits when studied in isolation, but their combination with other elements of the pathway is thought to have a synergistic effect. In this review, most impactful ERAS elements seemed to be patient education, NSAIDs with minimization of opioid use, local anesthesia, thromboprophylaxis, antibiotic prophylaxis, urinary catheters and drainage avoidance or removal within 24 h after surgery, TXA use and early mobilization within 24 h after surgery. The combined effects of these interventions have been shown to improve patient recovery with shorter LOS and decreases in hospital infections, complications, readmission rates and pain scores, with an increase in patients’ satisfaction due also to their active role and commitment. These aspects also lead to total cost savings, which accompany streamlined and less invasive methods. In this context, it is important to underline that, to date, ERAS costs have been estimated only in studies on THA and TKH surgery, and all indicated a reduction in medical costs compared with standard care with a prolonged LOS [29,66,188]. A recent study by Jansen et al. [195] also conduct a full economic evaluation with a cost-effectiveness analysis by using functional outcomes, LOS, thromboembolic complications, healthcare costs, and quality of life in TKA patients 12 months after surgery. Results showed a mean reduction in costs of EUR 268 per patient in favor of ERAS protocols, mostly due to the shorter LOS, which resulted in lower costs associated with nursing staff [195]. However, in general, and also in view of these cost analyses, it is difficult to extrapolate those elements that are less influential than others, also considering that good-quality data were not always available; thus, no recommendation can currently be made because either equipoise exists or there is a paucity of evidence. Stronger recommendations could be obtained from the 24 RCTs examined in this review, one on elbow, one on spine and 22 on hip and/or knee replacement surgery. In these RCTs, patient education and pre-emptive anesthetics and analgesics were the main pre-operative ERAS elements. A preoperative ERAS element of key importance little considered in these studies was the nutritional status [196]. in only two RCTs, it was reported that carbohydrate loading with a clear carbohydrate liquid 2 h prior to surgery was used in order to present the patient to surgery in a metabolically fed state leading to less postoperative protein loss and preservation of muscle mass [196]. This is probably due to the fact that this ERAS element requires special attention for those patients with specific comorbidities, such as obesity and diabetes, pathological conditions more common in aged patients [196]. Considering the intra-operative elements in the 24 RCTs, neuraxial anesthesia was frequently preferred to general anesthesia as well as multimodal analgesia, TXA use and urinary catheters and drainage avoidance or removal within 24 h after surgery. Although normothermia has been considered part of the anesthetic management in ERAS programs, no RCTs considered this aspect [3,196,197]. Hypothermia is common in patients who have undergone orthopedic surgery and may increase infection, coagulopathy, blood transfusion rate, cardiovascular complications, and opioid need, which may adversely affect the postoperative outcome [196]. Finally, in the post-operative phase, the main ERAS elements used in this RCT were early mobilization, opioid-sparing multimodal analgesia and thromboprophylaxis. Additoinally, as a post-operative element, no studies investigated the direct relationship between postoperative nutritional supplementation and accelerating the achievement of discharge criteria. However, encouraging patients to eat and drink as soon as possible is considered an essential component of the ERAS protocol, as returning to normal food intake can help patients return to normal behavior [3,196,197]. Considering all of these aspects of ERAS in orthopedic surgery, more investigations are mandatory to adapt and/or adjust several elements of the protocol [195]. Recently, under the impetus of the ERAS^®^ society, a multidisciplinary guideline development group was constituted by bringing together international experts involved in the practice of ERAS in spine surgery. This group identified 22 ERAS items specifically for lumbar fusion [197]. However, ERAS recommendations/guidelines also for other spinal procedures, cervical spine surgery, anterior or combined approaches, complex deformities, scoliosis, etc., and other orthopedic specialties are necessary.

Other critical key points to consider are whether further advances and implementations can be made to further reduce the risk of complications and, as the global trend is to shift to outpatient surgery, whether such orthopedic ERAS protocols can be performed on an ambulatory or semi-ambulatory basis without any increased risk of morbidity or cardiopulmonary and thromboembolic complications, as well as cognitive dysfunctions, especially in geriatric patients that have specific needs for rehabilitation. Last but not least, another important factor that emerged from the analyzed studies is the need for a unique, well-defined and updated guideline in every step and, importantly, a coordinated interaction between all the subjects involved, beginning from the very first ambulance’s intervention to the patient’s call. Based on these open questions, rigorous RCTs may serve to provide robust evidence and establish the efficacy of enhanced-recovery programs for particular patient populations and procedures within orthopedic surgery.

### 4.1. Limitation and Strengths

A methodological limitation of this review is correlated with the quality of the studies that were included. Most of these studies were retrospective studies, which are more likely subjected to biases than prospective randomized controlled trials. As highlighted by the quality assessment conducted, the moderate and weak scores were mainly associated with lack of a sample size justification and lack of blinded assessor or other potential confounding variables that could limit the validity of the review’s conclusions. On the other hand, to overcome these potential biases, the strength of this review stands in the development of an explicit and well-designed research protocol centered on a researchable and clinically relevant question that provide a clear description of the eligibility criteria such as population, intervention and outcomes of interest, the definition of explicit but also broad inclusion and exclusion criteria as well as the selection process. All of these methodological aspects were focused on extracting the best available evidence relevant to the review question. Additionally, as a patient-centered approach and evidence-based intervention, safety aspects following ERAS include morbidity and mortality, the first in the form of complications and readmissions. To the authors’ best knowledge, no disadvantages specifically related to the ERAS protocol in orthopedic surgery have been reported in the literature analyzed. However, several potential disadvantages should be assessed, such as the most demanding preoperative phase for the healthcare professional and for the patient, a phase that requires continuous multidisciplinary communication and collaboration. Furthermore, it would be essential to evaluate the real cost-effectiveness of ERAS protocol, examining and balancing the costs of all additional interventions with the specific patient advantages. Finally, the degree of independence of patients and the satisfaction associated with the shorter hospital stay should be analyzed in greater detail.

### 4.2. Future Prospects

Future studies focused on the elements of ERAS specific to orthopedic interventions, in particular for elbow, thorax and spine, may serve to optimize the protocol. Another critical key point to consider is whether further advances and implementations can be made to reduce even more the risk of complications and, as the global trend is to shift to outpatient surgery, whether such orthopedic ERAS protocols can be performed on an ambulatory or semi-ambulatory basis without any increased risk of morbidity or cardiopulmonary and thromboembolic complications, as well as cognitive dysfunctions, especially in geriatric patients that have specific needs for rehabilitation. Finally, another important factor that emerged from the analyzed studies is the need for a unique, well-defined and updated guideline in every step and, importantly, a coordinated interaction between all of the subjects involved, beginning from the very first ambulance’s intervention to the patient’s call. Based on these open questions, rigorous RCTs may serve to provide robust evidence and establish the efficacy of ERAS programs for particular patient populations and procedures within orthopedic surgery.

## Figures and Tables

**Figure 1 jcm-11-04222-f001:**
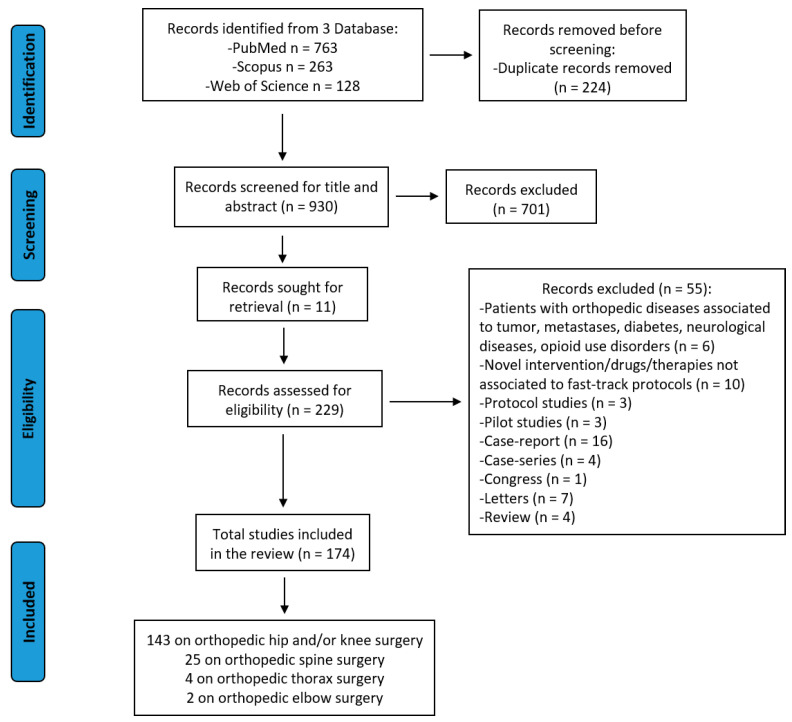
PRISMA 2020 flow diagram for the selection of studies.

**Figure 2 jcm-11-04222-f002:**
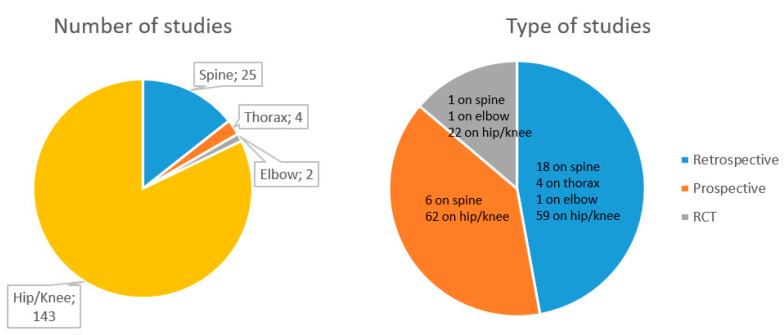
ERAS study characteristics, numbers, and types.

**Figure 3 jcm-11-04222-f003:**
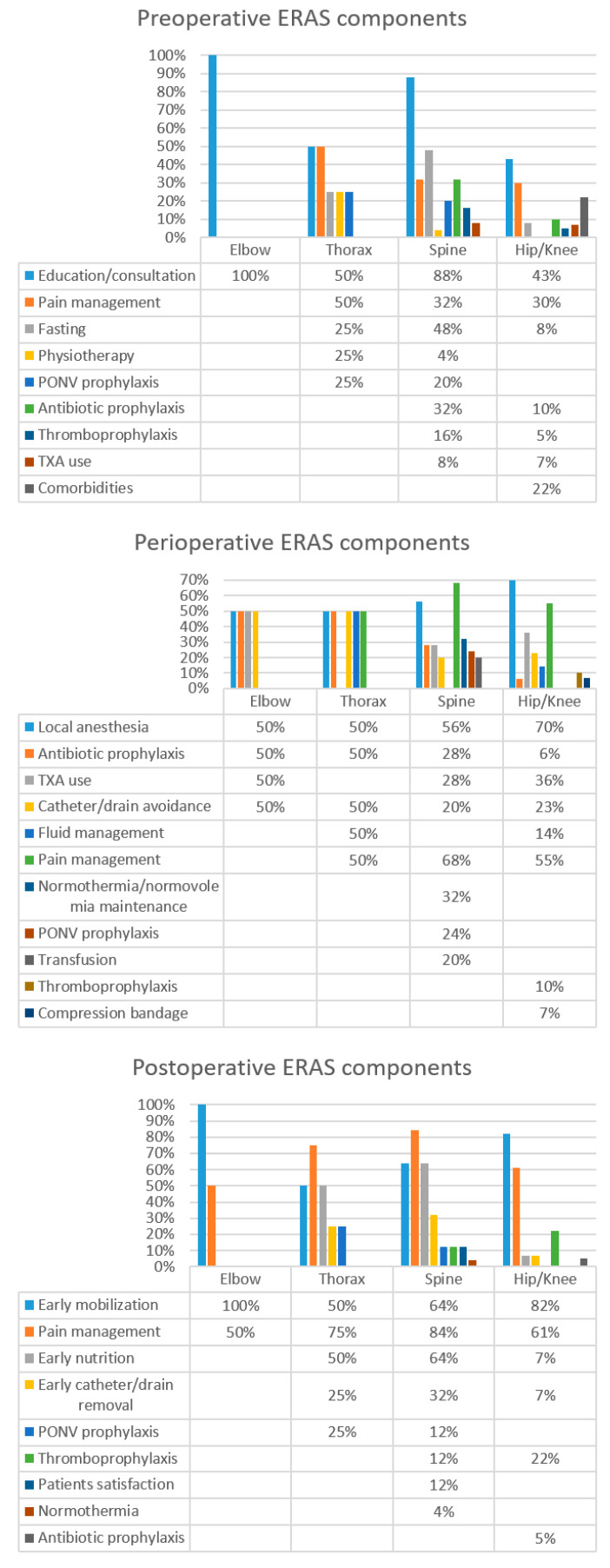
Pre-, peri- and postoperative elements of ERAS procedures.

## Data Availability

Not applicable.

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
