# Peer review of "Key Components, Current Practice and Clinical Outcomes of ERAS Programs in Patients Undergoing Orthopedic Surgery: A Systematic Review"

_jcm, 2022, doi:10.3390/jcm11144222_

Round 1

Reviewer 1 Report

Thank you for the opportunity to review the systematic review. The topic fast-track or enhanced recovery is an important area. The area has political attention as well, because it might be a part of the solution of the challenges the different health care systems have.

The review aims to address all specialities of orthopedic surgery, where as earlier published reviews only address for exampel THA. This is a strength of this review, but i leads to difficulties of reporting the results in an understandable manner. The manuscript needs more work in order to make it more understandable.

Title:

The title addresses only the fist part of your research question, it is maybe possible to integrate the clinical outcome as well.

Abstract:

The method of the systematic review is lacking in the abstract. It is crucial to the reader to understand the systematic method used. The authors should rewrite the abstract, in order to incorporate the method into it. In order to address the length of the abstract, there is a need of shorten and clarify the background and results part.

Introduction:

Matherials and Methods:

The review is not registered at PROSPERO. Why did you choose not to register the review in advance, as it will strengthen the review and methodology. The authors claim in strength and limitation section their explicit and well-designed research protocol - the registration could have helped there.

Line 92-98:

There is a lack of a reference to the PICOS framework. 

Line 99-101:

The focused question is not represented in the title. As well is the aim of the introduction wider then the focused question. Why is the measurement not present here, or vice versa.

Line 103:

How define you hypothetical fast-track designs? As you describe in the introduction, there is no standard. Please explain and clarify.

Line 106:

Why excluding different surgical apporaches within a fast-track protocol. You aim to cover all orthopedic surgery with the review. Please explain and clarify.

Line 138-142:

You describe that a fourth reviewer made the final desicion. Meanwhile you describe only 2 reviewers. I anticipate that you mean the foruth reviewer (MF) from the selection process. Please correct and clarify.

Results:

Line 160:

Search strategy is shown in your supplemental material. Search result may better fit.

Line 193-195:

The authors should report all details per specialty. As they began with, and keep the intial order of reporting.

Line 195-197:

The authors introduced an order of the specialties. Please hold the order throughout the reporting of the results, as the readability is higher.

Line 197-207:

Again a new order of the specialties, making it unnecessary diffucult reading the manuscript.

Section 3.3.2:

The section needs generally rewriting. The manner of reporting is making it difficult to follow as a reader. Kombining different specialties together in one sentence an kombine others in the next sentence. Due to the complexity of this outcome, it would be to prefer, to report the outcome by specialty. (spine, thorax, elbow and THA/TKA as introduced very first in the result section). It might be necessary to go through the subsections by specialty to make it more comparable.

Line 233:

Comorbities assessment is lacking in figure 3

Line 249-256:

Normothermia/normovolemia maintenance, postoperative nausea and vomiting profylaxis, transfusion control, thrombosisprofylaxis and compression bandage use is lacking in figure 3.

Line 269-272:

Patients satisfaction survey and normothermia are lacking in figure 3.

Figure 3:

Is helping the reader to understand the section, but there i lacking the above mentioned subjects.

Section 3.4:

Throughout the section the reporting i grouping and regrouping the specialties. Try to seperate the section by specialty or keep the started systematic through the whole section.

Line 274-297:

There is i mixing of simply reporting outcomes, with concrete results, specially concerning LOS.

Line 298-342:

Discussion:

There is a lack of references for comparison in the discussion. This will give your findings a better relation. There should be a discussion, if there is the possibility of generalization out of your findings. There should be addressed the few articles in some specialties and the quality scores of the articles.

Limitations and strengths:

In the reporting the authors are not relate the results to the quality assessement, which is a weakness. An other could be not to use articles with a weak score. The authors should argue further for their choices.

Author Response

Comments and Suggestions for Authors

-Thank you for the opportunity to review the systematic review. The topic fast-track or enhanced recovery is an important area. The area has political attention as well, because it might be a part of the solution of the challenges the different health care systems have.

The review aims to address all specialities of orthopedic surgery, where as earlier published reviews only address for exampel THA. This is a strength of this review, but i leads to difficulties of reporting the results in an understandable manner. The manuscript needs more work in order to make it more understandable.

After the reviewer recommendations we reported the results more clearly by answering to all the reviewer's comments and including all suggestions.

-Title: The title addresses only the fist part of your research question, it is maybe possible to integrate the clinical outcome as well.

As suggested, we modified the title of this review as follow: “Key components, current practice and clinical outcomes of fast-track programs in patients undergoing orthopedic surgery: a systematic review”

-Abstract: The method of the systematic review is lacking in the abstract. It is crucial to the reader to understand the systematic method used. The authors should rewrite the abstract, in order to incorporate the method into it. In order to address the length of the abstract, there is a need of shorten and clarify the background and results part.

As suggested, we modified the abstract including the systematic method used and shortening and clarifying the background and results sections. (Lines 16-38)

Matherials and Methods

-The review is not registered at PROSPERO. Why did you choose not to register the review in advance, as it will strengthen the review and methodology. The authors claim in strength and limitation section their explicit and well-designed research protocol - the registration could have helped there.

We agree with the reviewer, and we are in the process of registering the review in PROSPERO since we are aware that the registration of the review promotes transparency, helps reduce potential for bias and serves to avoid unintended duplication of reviews.

-Line 92-98: There is a lack of a reference to the PICOS framework.

We added the reference on PICOS framework (reference number 17) (Line 99).

-Line 99-101: The focused question is not represented in the title. As well is the aim of the introduction wider then the focused question. Why is the measurement not present here, or vice versa.

After the review suggestion, we modified the title of the review including the focused question. In addition, we also modified the Introduction section to better underline the aim of the review (Line 94).

-Line 103: How define you hypothetical fast-track designs? As you describe in the introduction, there is no standard. Please explain and clarify.

We apologize to the reviewer for the misunderstanding. By the term "hypothetical" we meant the studies in which the use of a fast-track protocol was declared but which then did not follow any of the indications of a fast-track protocol. We now clarify the sentence in the manuscript. (Line 111).

-Line 106: Why excluding different surgical apporaches within a fast-track protocol. You aim to cover all orthopedic surgery with the review. Please explain and clarify.

We the sentence “different surgical approaches within a fast-track protocol” we meant different surgeries, also different from orthopedics, within a fast-track protocol. We now clarify the sentence. (Line 114).

-Line 138-142: You describe that a fourth reviewer made the final desicion. Meanwhile you describe only 2 reviewers. I anticipate that you mean the foruth reviewer (MF) from the selection process. Please correct and clarify.

We corrected the sentence. (Line 150).

Results:

-Line 160: Search strategy is shown in your supplemental material. Search result may better fit.

We corrected the sentence as suggested by the reviewer. (Line 173).

-Line 193-195: The authors should report all details per specialty. As they began with, and keep the intial order of reporting.

After the reviewer we always maintained the same order, i.e. elbow, thorax, spine, hip/knee. In Figure 2 we reported all the details for the different orthopedic specialties.

-Line 195-197: The authors introduced an order of the specialties. Please hold the order throughout the reporting of the results, as the readability is higher.

After the reviewer we always maintained the same order, i.e. elbow, thorax, spine, hip/knee.

-Line 197-207: Again a new order of the specialties, making it unnecessary diffucult reading the manuscript.

As previously reported, we corrected the manuscript maintaining always the same order, i.e. elbow, thorax, spine, hip/knee.

-Section 3.3.2: The section needs generally rewriting. The manner of reporting is making it difficult to follow as a reader. Kombining different specialties together in one sentence an kombine others in the next sentence. Due to the complexity of this outcome, it would be to prefer, to report the outcome by specialty. (spine, thorax, elbow and THA/TKA as introduced very first in the result section). It might be necessary to go through the subsections by specialty to make it more comparable.

As suggested by the reviewer we rewrite the Section 3.3.2 and divided the different subsection by specialty (i.e. elbow, thorax, spine and THA/TKA).

 -Line 233: Comorbities assessment is lacking in figure 3

We have modified Figure 3 as suggested by the reviewer.

-Line 249-256: Normothermia/normovolemia maintenance, postoperative nausea and vomiting profylaxis, transfusion control, thrombosisprofylaxis and compression bandage use is lacking in figure 3.

We have modified Figure 3 as suggested by the reviewer.

-Line 269-272: Patients satisfaction survey and normothermia are lacking in figure 3.

We have modified Figure 3 as suggested by the reviewer.

-Figure 3: Is helping the reader to understand the section, but there i lacking the above mentioned subjects.

We have modified Figure 3 as suggested by the reviewer.

-Section 3.4: Throughout the section the reporting i grouping and regrouping the specialties. Try to seperate the section by specialty or keep the started systematic through the whole section.

The initial part of the Section 3.4 is divided for the different specialties. The second part of the section 3.4 is a general part where we tried to make a comparison between the different orthopedic specialties. Following your comment, we have tried to make this part clearer and more fluent in order to be more understandable to the reader.

-Line 274-297: There is i mixing of simply reporting outcomes, with concrete results, specially concerning LOS.

After the reviewer suggestion, we modified the Section 3.4 where data on LOS were reported. We tried to make these data clearer and more fluent.

Discussion:

-There is a lack of references for comparison in the discussion. This will give your findings a better relation. There should be a discussion, if there is the possibility of generalization out of your findings. There should be addressed the few articles in some specialties and the quality scores of the articles.

As suggested by the reviewer we modified and added several references to the discussion section. (Lines 413-484).  In addition, we referred to and discussed more in detail the RCT studies, as these will be the definitely most proper source for stating anything on the impact of the fast-track program per se (Lines 443-470).  

Limitations and strengths:

-In the reporting the authors are not relate the results to the quality assessement, which is a weakness. Another could be not to use articles with a weak score. The authors should argue further for their choices.

To overcome the weakness of several of the analyzed studies, in this review we developed an explicit and well-designed research protocol centered on a researchable, and clinically relevant question which provide a clear description of the eligibility criteria such as population, intervention and outcomes of interest, the definition of explicit but also broad inclusion and exclusion criteria as well as the selection process. All these methodological aspects were focused on extracting the best available evidence relevant to the review question. Obviously, this does not eliminate the limitations linked to the studies analyzed but we think that eliminating the weaker articles would still have limited our data and the vision of what is actually present on the fast-track today.

Reviewer 2 Report

This is a very interesting topic. This work can be of help to other health care providers to create a Fast-track system in their hospitals.

Because of the subject matter, it is a bit hard to read. But it is not a topic that can be summarised in a simple way.

I only recommend adding all references in the article, not only in the Supplementary Material. 

From my point of view, the authors have done serious, orderly and publishable work.

Author Response

Comments and Suggestions for Authors

This is a very interesting topic. This work can be of help to other health care providers to create a Fast-track system in their hospitals.

Because of the subject matter, it is a bit hard to read. But it is not a topic that can be summarised in a simple way.

I only recommend adding all references in the article, not only in the Supplementary Material.

From my point of view, the authors have done serious, orderly and publishable work.

We thank the reviewer and following her/his suggestions we have tried to make the reading of the manuscript clearer and more fluent, and we have added references in the Section ‘materials and methods’ and in the ‘discussion’.

Reviewer 3 Report

Salamanna and co-workers have done an extensive and methodological sound work in retrieving publications on fast-track programs in orthopaedic in-patients during a ten years period. They provide a comprehensive listing of key components in such programs (pre-, per- and post-operatively) in text and table, and a descriptive report on some outcome parameters. 

Comments:

11)      As the author discuss, “Fast-track” is also named “Enhanced Recovery After Surgery” (ERAS), which is a better and more precise description of the prioritized goals of the efforts undertaken. Although faster recovery and faster perioperative handling is one of the outcomes, “speed” per se is not an important outcome for the patient or the health care workers, although there are some positive economic implications unless new costs are added from other issues. Thus, I will like the authors to focus on the enhanced recovery as a better term, because this will emphasize the goals of better-perceived quality for the patient as well as better functional outcomes without the expense of more side-effects or complications. The term “fast-track surgery” may also be misinterpreted in US literature, as it mean to bypass the recovery unit after surgery, ie. taking the patient directly from the OR to the ward.

22)      A major problem with Fast-track/ERAS protocols is to know which of the many elements really have an impact, thus if any of these seemingly promising measures may be skipped without resulting in inferior results. A discussion and references on this issue is missing as a part of the present manuscript. Thus we do not know which of the measures have a sound evidence based rationale and which are included because “some” may consider them potentially useful. This is important, as including non-documented issues in a protocol will imply more costs, more time-consume and potential risks of non-justified side-effects. Some of the recent ERAS works in other surgical areas have actually focused on which measures to skip, in order to further improve outcomes and cost-efficacy.

33)      Only 1 out of six studies were RCTs and these should be referred to and discussed more in detail, as these will be the definitely most proper source for stating anything on the impact of the ERAS/fast-track program per se.  This is because most of the nice measures and drugs used in an ERAS/fast-track protocol  (and subsequent better results!) could be a part of any department policy over time, and not necessarily designated to the ERAS/fast-track concept. Even with a prospective cohort study the participants will know they are studied for the purpose of improvement, which eventually will take place irrespectively of whether a strict protocol is used or not.

44)      The outcomes part is the most interesting part of this study and should be expended and improved in some aspects:

a)       The RCT should be reported separately from the retrospective cohort with the prospective cohort as a third group.

b)      A table of the results should be given, with a numerical designation on how many studies who are significantly positive, neutral and negative for each outcome.  For instance, for thorax orthopaedic surgery a “ reduction in intraoperative time (25%)”  is reported.  This should not be interpreted (as it may be by quick glance!) to a 25% reduction in intraoperative time, but as 1 out of four studies showing significant time reduction. In a table this will be more clear and easy to read, 1 study gave reduction,  3 did not (and were all 3 neutral or was any with increase in time consume?)

c)       The outcomes are a non-systematic mixture of mostly surrogate outcomes and some patient relevant ones. The primary outcome of shorter length of stay is mainly of interest to the budget providers, but may be a surrogate for patients being more rapidly and better recovered as well. Similarly, with opioid reduction which are non-important unless the side-effects are reduced and/or the functions improved at the same time. Post-operative pain scores, non-increase in side-effects, better range of motion, less readmissions, early resumption of eating, better patient satisfaction are all relevant and nice outcomes, but these come from the total study cohort of 174 studies. These will have a lot of confounders, the 24 RCTs should be used to really spot these benefits in an evidence based way.

55)      The authors do not report or discuss potentially negative consequences of  ERAS/fast-track: The preoperative period will be more labour-intensive and potentially stressed (for patients and workers), the costs of all extra drugs and measures may definitely add to total costs and should be balanced against a potential cost reduction of shorter (and less relaxed?) stay. Also, some patients may not be satisfied by short stay and rushed pathways of hospital care, was this looked for or reported in any of the studies?

66)      The authors should discuss the limitations of not including studies with patients having concomitant pathological conditions, as these patients will be of increasing importance in the increasingly elderly population in a mixed orthopedic surgical practice. Is the list of conditions, which resulted in exclusion complete? Is dementia a part of “neurological disorders” and were demented patients excluded from the study?

Author Response

1) As the author discuss, “Fast-track” is also named “Enhanced Recovery After Surgery” (ERAS), which is a better and more precise description of the prioritized goals of the efforts undertaken. Although faster recovery and faster perioperative handling is one of the outcomes, “speed” per se is not an important outcome for the patient or the health care workers, although there are some positive economic implications unless new costs are added from other issues. Thus, I will like the authors to focus on the enhanced recovery as a better term, because this will emphasize the goals of better-perceived quality for the patient as well as better functional outcomes without the expense of more side-effects or complications. The term “fast-track surgery” may also be misinterpreted in US literature, as it mean to bypass the recovery unit after surgery, ie. taking the patient directly from the OR to the ward.

We thank the reviewer very much for the comment and as suggested we changed the term ‘fast-track’ to the term ‘ERAS’, thus, so as not to be misinterpreted.

2) A major problem with Fast-track/ERAS protocols is to know which of the many elements really have an impact, thus if any of these seemingly promising measures may be skipped without resulting in inferior results. A discussion and references on this issue is missing as a part of the present manuscript. Thus we do not know which of the measures have a sound evidence based rationale and which are included because “some” may consider them potentially useful. This is important, as including non-documented issues in a protocol will imply more costs, more time-consume and potential risks of non-justified side-effects. Some of the recent ERAS works in other surgical areas have actually focused on which measures to skip, in order to further improve outcomes and cost-efficacy.

As suggested by the reviewer we added a discussion and several references on specific elements that really could have an impact in ERAS protocols for orthopaedic surgery (Lines 413-484). However, considering that good-quality data were not always available in the studies analysed in this review and thus it is difficult to extrapolate specific ERAS elements that are less and/or more influential than others, we tried to extrapolate these data from the 24 RCTs examined in this review. (Lines 443-470).  

3)  Only 1 out of six studies were RCTs and these should be referred to and discussed more in detail, as these will be the definitely most proper source for stating anything on the impact of the ERAS/fast-track program per se.  This is because most of the nice measures and drugs used in an ERAS/fast-track protocol  (and subsequent better results!) could be a part of any department policy over time, and not necessarily designated to the ERAS/fast-track concept. Even with a prospective cohort study the participants will know they are studied for the purpose of improvement, which eventually will take place irrespectively of whether a strict protocol is used or not.

As suggested, we discussed the RCTs data separately in the discussion section (Lines 443-470).  

4) The outcomes part is the most interesting part of this study and should be expended and improved in some aspects:

a)       The RCT should be reported separately from the retrospective cohort with the prospective cohort as a third group.

As suggested, we discussed the RCTs data separately in the discussion section. (Lines 443-470). 

b)    A table of the results should be given, with a numerical designation on how many studies who are significantly positive, neutral and negative for each outcome.  For instance, for thorax orthopaedic surgery a “ reduction in intraoperative time (25%)”  is reported.  This should not be interpreted (as it may be by quick glance!) to a 25% reduction in intraoperative time, but as 1 out of four studies showing significant time reduction. In a table this will be more clear and easy to read, 1 study gave reduction,  3 did not (and were all 3 neutral or was any with increase in time consume?)

We added a new table (Table 8) where we reported with a numerical designation how many studies are positive, neutral and negative for each specific outcome.

 c)       The outcomes are a non-systematic mixture of mostly surrogate outcomes and some patient relevant ones. The primary outcome of shorter length of stay is mainly of interest to the budget providers, but may be a surrogate for patients being more rapidly and better recovered as well. Similarly, with opioid reduction which are non-important unless the side-effects are reduced and/or the functions improved at the same time. Post-operative pain scores, non-increase in side-effects, better range of motion, less readmissions, early resumption of eating, better patient satisfaction are all relevant and nice outcomes, but these come from the total study cohort of 174 studies. These will have a lot of confounders, the 24 RCTs should be used to really spot these benefits in an evidence based way.

 As suggested, we discussed the RCTs data separately in the discussion section. (Lines 443-470).  

5)  The authors do not report or discuss potentially negative consequences of  ERAS/fast-track: The preoperative period will be more labour-intensive and potentially stressed (for patients and workers), the costs of all extra drugs and measures may definitely add to total costs and should be balanced against a potential cost reduction of shorter (and less relaxed?) stay. Also, some patients may not be satisfied by short stay and rushed pathways of hospital care, was this looked for or reported in any of the studies?

As suggested, we discussed the potential negative consequences of ERAS protocol in the discussion section. (Lines 515-521). 

 6) The authors should discuss the limitations of not including studies with patients having concomitant pathological conditions, as these patients will be of increasing importance in the increasingly elderly population in a mixed orthopedic surgical practice. Is the list of conditions, which resulted in exclusion complete? Is dementia a part of “neurological disorders” and were demented patients excluded from the study?

We apologize with the reviewer but concerning the studies on patients with neurological diseases we intended rare neurological diseases, such as pediatric ataxia, megalencephalic leukoencephalopathy with subcortical cysts, Creutzfeldt-Jakob disease, amyotrophic lateral sclerosis. We corrected the sentence in the material and method section. (Line 113). 

Round 2

Reviewer 1 Report

Thank you for the opportunity to review the revised version of the systematic review. As mentioned before the topic of the review is very actual and important.

The paper improved substantially, due to the revision. The readability und understandabilty i now very good.

I have only a few minor comments to the improved and revised paper.

Line 272: TXA as acronym has no introduction, please insert the full term first. 

Figure 3: Antibiotic prophylaxis is in the text under hip/knee and not spine (also in the first version). Please adjust.

Line 315-316: Correct the order from highest to lowest, as you did consistent before. Pain management and rehabilitation/physiotherapy should change place.

Line 318-319: As above, correct the order from highest to lowest. Change the place pain management and rehabilitation/physiotherapy.

Line 325-326: As stated above correct the order. Thromboprophylaxis should be before early nutrition.

Line 520: Typos: degree of independence of patients instead.

Author Response

We thank the reviewer for the comment.

- As suggested by the reviewer, we added the full term of TXA (line 249), adjusted the Figure 3, corrected the order from highest to lowest (lines 290-292, 294-295, 301), and corrected "degree of independence of patients" (line 472)

Reviewer 3 Report

I think the authors have done a huge job, and that the present version is interesting for the readers.

Author Response

We thank the reviewer for the comment